# LEARNING TO DEFENSE BY LEARNING TO ATTACK

**Haoming Jiang**\*, **Zhehui Chen**\*, **Yuyang Shi, Tuo Zhao**
H. Milton Stewart School of Industrial and Systems Engineering
Georgia Institute of Technology
Atlanta, GA, USA
{jianghm,zhchen,yyshi,tourzhao}@gatech.edu

**Bo Dai**
Google Brain
Mountain View, CA, USA
bodai@google.com

## ABSTRACT

Adversarial training provides a principled approach for training robust neural networks. From an optimization perspective, the adversarial training is essentially solving a minmax robust optimization problem. The outer minimization is trying to learn a robust classifier, while the inner maximization is trying to generate adversarial samples. Unfortunately, such a minmax problem is very difficult to solve due to the lack of convex-concave structure. This work proposes a new adversarial training method based on a general learning-to-learn framework. Specifically, instead of applying the existing hand-designed algorithms for the inner problem, we learn an optimizer, which is parametrized as a convolutional neural network. At the same time, a robust classifier is learned to defense the adversarial attack generated by the learned optimizer. From the perspective of generative learning, our proposed method can be viewed as learning a deep generative model for generating adversarial samples, which is adaptive to the robust classification. Our experiments demonstrate that our proposed method significantly outperforms existing adversarial training methods on CIFAR-10 and CIFAR-100 datasets.

## 1 INTRODUCTION

This decade has witnessed great breakthroughs in deep learning in a variety of applications, such as computer vision (Taigman et al., 2014; Girshick et al., 2014; He et al., 2016; Liu et al., 2017). Recent studies (Szegedy et al., 2013), however, show that most of these deep learning models are very vulnerable to adversarial attacks. Specifically, by injecting a small perturbation to a normal sample, one can obtain an adversarial example. Although the adversarial example is semantically indistinguishable from the normal one, it can severely fool deep learning models and undermine the security of deep learning, causing reliability problems in autonomous driving, biometric authentication, etc.

Researchers have devoted many efforts to studying efficient adversarial attack and defense (Szegedy et al., 2013; Goodfellow et al., 2014b; Nguyen et al., 2015; Zheng et al., 2016; Madry et al., 2017; Carlini & Wagner, 2017). There is a growing body of work on generating successful adversarial examples, e.g., fast gradient sign method (FGSM, Goodfellow et al. (2014b)), projected gradient method (PGM, Kurakin et al. (2016)), Carlini-Wagner (CW, Paszke et al. (2017)) etc. As for defense, Goodfellow et al. (2014b) first propose to robustify the network by adversarial training, which augments the training data with adversarial examples and still requires the network to output the correct label. Further, Madry et al. (2017) formalize the adversarial training as the following minmax optimization problem:

$$\min_\theta \quad \frac{1}{n} \sum_{i=1}^n \left[ \max_{\delta_i \in \mathcal{B}} \ell(f(x_i + \delta_i; \theta), y_i) \right], \tag{1}$$

where $\{(x_i, y_i)\}_{i=1}^n \subset \mathbb{R}^d \times \mathcal{Y}$ are $n$ pairs of input feature and the corresponding label, $\ell$ denotes a loss function, $f(\cdot; \theta)$ denotes the neural network with parameter $\theta$, and $\delta_i \in \mathcal{B}$ denotes the perturbation for $x_i$ in $\mathcal{B}$. The existing literature on optimization also refers to $\theta$ as the primal variable and $\delta_i$'s as the dual variables. Different from the well-studied convex-concave problem[1], problem equation 1 is very challenging. Since $\ell$ is nonconvex in $\theta$ and nonconcave in $\delta$, there exist many equilibria. The majority of them are unstable. In the existing optimization literature, there is no algorithm to converge to a stable equilibrium with theoretical guarantees. Empirically, the existing primal-dual algorithms perform poorly for solving equation 1.

---

[1]Loss function $\ell(\theta; \delta)$ is convex in primal variable $\theta$ and concave in dual variable $\delta$.

The minmax formulation in equation 1 naturally provides us with a unified perspective on prior works of adversarial training. Such a minmax problem consists of two optimization problems, an inner maximization problem and an outer minimization problem: The inner problem targets on finding an optimal attack for a given data point $(x, y)$ that maximizes the loss, which essentially is the adversarial attack; The outer problem aims to find a $\theta$ so that the loss given by the inner problem is minimized. For solving equation 1, Goodfellow et al. (2014b) propose to use FGSM to solve the inner problem. Madry et al. (2017) further suggest to solve the inner problem by PGM and obtain a result better than FGSM, since FGSM essentially is one iteration PGM. PGM, however, still does not guarantee to find the optimal solution of the inner problem, due to the nonconcavity of the inner problem. Furthermore, PGM training does not obtain a stable equilibrium of problem equation 1. Moreover, adversarial training needs to find a $\delta_i$ for each $(x_i, y_i)$, thus the dimension of the overall search space for all data is substantial, which makes the computation unaffordable. Besides, existing methods, e.g., FGSM and PGM, suffer from the gradient vanishing in backpropagation, which makes the gradient uninformative and slows down the training procedure.

Without much prior knowledge (well-studied structure) of the loss function, the hand-designed methods are not guaranteed to generate good perturbations and achieve a good performance. Instead, for solving equation 1, we propose a new learning-to-learn (L2L) framework that provides a more efficient and flexible way to generate strong perturbations for adversarial training. Specifically, we parameterize the optimizer of the inner maximization problem by a convolutional neural network (CNN) denoted by $g(\mathcal{A}(x, y, \theta); \phi)$, where $\mathcal{A}(x, y, \theta)$ denotes an operator yielding the input of the optimizer. We also call the optimizer as the attacker network. Since the neural network is very powerful in function approximation, such a parameterization ensures that our attacker network $g$ is able to yield strong adversarial perturbations. Under our framework, instead of directly solving $\delta_i$, we update the parameter $\phi$ of the attacker network $g$. Our training procedure then becomes updating the parameters of two neural networks, which is very similar to Generative Adversarial Network (GAN, Goodfellow et al. (2014a)).

Different from the hand-designed methods computing the adversarial perturbation for each individual sample using gradients from backpropagation, our methods generate the adversarial perturbations for all samples through the shared attacker network $g$. This enables the attacker network to learn potential common structures of the adversarial perturbations for all samples. Therefore, our method is capable of yielding strong perturbations and accelerating the training process. Furthermore, the L2L framework is very flexible. We can either choose different $\mathcal{A}(x, y, \theta)$ as the input of the attacker, or use different attacker network architectures. For example, we can include gradient information in $\mathcal{A}(x, y, \theta)$ and use a recurrent neural network (RNN) to mimic multi-step gradient-type algorithms. In this paper, we mainly consider three attacker networks: (1) Naive Attacker Network with $\mathcal{A}(x, y, \theta) = x$; (2) Gradient Attacker Network with with $\mathcal{A}(x, y, \theta) = [x, \nabla_x \ell]$; (3) Multi-Step Gradient Attacker Network with $\mathcal{A}(x, y, \theta) = [x, \nabla_x \ell]$. The last two attacker networks may potentially capture the high order information to help the adversarial training. Instead of simply computing the high order information with finite difference approximation or multiple gradients, by parameterizing the algorithm by a neural network, our proposed methods can capture this information in a much smarter way (Finn et al., 2017). Our experiments demonstrate that our proposed methods significantly outperform existing adversarial training methods, e.g., FGSM training, over SVHN, CIFAR-10, and CIFAR-100 datasets (Netzer et al., 2011; Krizhevsky & Hinton, 2009).

The research on learning-to-learn has a long history (Schmidhuber, 1987; 1992; 1993; Younger et al., 2001; Hochreiter et al., 2001; Andrychowicz et al., 2016). The basic idea is that one first models the updating formula of complicated optimization algorithms in a parametric form, and then uses some simple algorithms, e.g., stochastic gradient algorithm to learn the parameter of the optimizer. Among existing works, Hochreiter et al. (2001) propose a system allowing the output of backpropagation from one network to feed into an additional learning network, with both networks trained jointly; Based on this, Andrychowicz et al. (2016) further show that the design of an optimization algorithm can be cast as a learning problem. Specifically, they use long short-term memory RNNs to model the algorithm and allow the RNNs to exploit structure in the problems of interest in an automatic way, which is undoubtedly one of the most popular methods for learning-to-learn.

However, there are two major drawbacks of the existing learning-to-learn methods: **(1)** It requires a large amount of datasets (or a large number of tasks in multi-task learning) to guarantee the learned optimizer to generalize, which significantly limits their applicability (most of the related works only consider the image encoding as the motivating application); **(2)** The number of layers/iterations in the

RNN for modeling algorithms cannot be very large so as to avoid significant computational burden in backpropagation.

Our contribution is that we fill the blank of the learning-to-learn framework in solving minmax problem, and our proposed methods do not suffer from the aforementioned drawbacks: **(1)** The attacker network $g$ with a different $\phi$ essentially generates a different task/dataset. Therefore, for adversarial training, we have sufficient enough tasks for learning-to-learn; **(2)** The inner problem does not need a large scale RNN, and we use a CNN or a length-two RNN (sequence of length 2) as our attacker network, which eases the computational issue.

Our work is closely related to GAN and dual-embedding (Goodfellow et al., 2014a; Dai et al., 2016), since from the perspective of generative learning, our proposed method can be viewed as learning a deep generative model for generating adversarial samples, which is adaptive to the robust classification. All these works focus on solving minmax problems and share some common ground. We will discuss these works in detail later.

**Notations**. Given $a \in \mathbb{R}$, we denote $(a)_+$ as $\max(a, 0)$. Given two vectors $x, y \in \mathbb{R}^d$, we denote $x_i$ as the $i$-th element of $x$, $||x||_\infty = \max_i |x_i|$ as the $\ell_\infty$-norm of $x$, and $x \circ y = [x_1 y_1, \cdots, x_d y_d]^\top$ as the element-wise product.

## 2 METHOD

This paper focuses on the $\ell_\infty$-norm attack. We define the $\ell_\infty$-ball with radius $\epsilon$ by $\mathcal{B}(\epsilon) = \{\delta \in \mathbb{R}^d : ||\delta||_\infty \le \epsilon\}$ and the corresponding projection as follows:

$$\Pi_{\mathcal{B}(\epsilon)}(\delta) = \text{sign}(\delta) \circ \max(|\delta|, \epsilon),$$

where $\text{sign}(\cdot)$ and $\max(\cdot, \cdot)$ are element-wise operators.

### 2.1 ADVERSARIAL TRAINING AS ROBUST OPTIMIZATION

The goal of adversarial training is to robustify neural networks. Recall that from a robust optimization perspective, given $n$ samples $\{(x_i, y_i)\}_{i=1}^n$, where $x_i$ is the $i$-th feature vector and $y_i$ is the corresponding label, the adversarial training is reformulated as minmax problem equation 1:

$$\min_\theta \quad \frac{1}{n} \sum_{i=1}^n \left[ \max_{\delta_i \in \mathcal{B}(\epsilon)} \ell(f(x_i + \delta_i; \theta), y_i) \right],$$

where $f$ denotes the network with parameter $\theta$, $\ell$ denotes a loss function, and $\epsilon$ is the maximum perturbation magnitude. The inner problem aims to find a perturbation $\delta_i$ of $x_i$ such that $x_i + \delta_i$ increases the value of loss function as much as possible; While the outer problem targets on decreasing the loss value with the perturbed data $x_i + \delta_i$. In the existing literature, the standard pipeline of adversarial training is shown in Algorithm 1.

Note that the loss function $\ell(f(x_i + \delta_i; \theta), y_i)$ is highly nonconcave in $\delta_i$. Therefore the step of generating adversarial perturbation $\delta_i$ in Algorithm 1 is intractable. In practice, this step in most adversarial training methods adopts hand-designed algorithms. For example, Kurakin et al. (2016) propose to solve the inner problem approximately by first order methods such as PGM. Specifically, PGM iteratively updates the adversarial perturbation by projected sign gradient ascent method for each sample: Given one sample $(x_i, y_i)$, at the $t$-th iteration, PGM takes

$$\delta_i^t \leftarrow \Pi_{\mathcal{B}(\epsilon)}\left(\delta_i^{t-1} + \eta \cdot \text{sign}\left(\nabla_x \ell(f(\widetilde{x}_i; \theta), y)\right)\right), \tag{2}$$

where $\widetilde{x}_i = x_i + \delta_i^{t-1}$, $\eta$ is the step size, $T$ is a pre-defined total number of iterations, $\delta_i^0 = 0$, $t = 1, \cdots, T$, and $\text{sign}(\cdot)$ is an element-wise operator. Finally PGM takes $\delta_i = \delta_i^T$. Note that FGSM essentially is an one-iteration version of PGM. Besides, some works adopt other optimization methods, such as momentum gradient method (Dong et al., 2018), and L-BFGS (Tabacof & Valle, 2016). However, except for FGSM, they all require numerous queries for gradients through backpropagation, which is computationally expensive.

### 2.2 LEARNING TO DEFENSE BY LEARNING TO ATTACK (L2L)

Since hand-designed methods do not perform well, we propose to learn an optimizer for the inner problem. Specifically, we parameterize the attacker by a CNN $g(\mathcal{A}(x, y, \theta); \phi)$, where the input of the network $g$, $\mathcal{A}(x, y, \theta)$, summaries the information of the data and the neural network $f(\cdot; \theta)$. We then convert problem equation 1 to a two stage optimization problem as follows:

$$\min_\theta \frac{1}{n} \sum_{i=1}^n \left[ \ell(f(x_i + g(\mathcal{A}(x_i, y_i, \theta); \phi^*); \theta), y_i) \right], \tag{3}$$

---

**Algorithm 1** *Standard pipeline of adversarial training*

---

**Input:** $\{(x_i, y_i)\}_{i=1}^{n}$: clean data, $\alpha$: step size, $N$: number of epochs, $\epsilon$: maximum perturbation magnitude.
**Return:** $\theta$: parameter of classifier network $f$.
**for** $t \leftarrow 1$ *to* $N$ **do**
    Sample a minibatch $\mathcal{M}_t$
    **for** $i$ *in* $\mathcal{M}_t$ **do**
        $\delta_i \leftarrow \arg\max_{\delta_i \in \mathcal{B}(\epsilon)} \ell(f(x_i + \delta_i; \theta), y_i)$
        Generate adversarial perturbation for $(x_i, y_i)$
    $\theta \leftarrow \theta - \alpha \frac{1}{|\mathcal{M}_t|} \sum_{i \in \mathcal{M}_t} \nabla_\theta \ell(f(x_i + \delta_i; \theta), y_i)$
    Update $\theta$ over adversarial data $\{(x_i + \delta_i, y_i)\}_{i \in \mathcal{M}_t}$

---

where $\phi^*$ is defined as the solution to the following optimization problem:

$$\phi^* \in \arg\max_\phi \frac{1}{n} \sum_{i=1}^{n} \ell(f(x_i + g(\mathcal{A}(x_i, y_i, \theta); \phi); \theta), y_i)$$
$$\text{subject to} \quad g(\mathcal{A}(x, y, \theta); \phi) \in \mathcal{B}(\epsilon).$$

Solving problem equation 3 naturally consists of two stages. In the first stage, the classifier $f$ aims to fit over all perturbed data; While in the second stage, given a certain classifier $f$ obtained in the first stage, the attacker network $g$ targets on generating optimal perturbations under constraints $\delta_i$'s $\in \mathcal{B}(\epsilon)$. Since $\delta_i = g(\mathcal{A}(x_i, y_i; \theta); \phi)$, the constraints can be simply handled by a $\tanh$ activation function in the last layer of the attacker network $g$. Specifically, because the magnitude of $\tanh$ output is bounded by 1, after we rescale the output by $\epsilon$, the output of the network $g$ automatically satisfies the constraints.

---

**Algorithm 2** *Learning-to-learn-based adversarial training with gradient attacker network*

---

**Input:** $\{(x_i, y_i)\}_{i=1}^{n}$: clean data, $\alpha_1, \alpha_2$: step sizes, $N$: number of epochs, $\epsilon$: maximum perturbation magnitude.
**Return:** $\theta$: parameter of classifier network $f$; $\phi$: parameter of attacker network $g$.
**for** $t \leftarrow 1$ *to* $N$ **do**
    Sample a minibatch $\mathcal{M}_t$
    **for** $i$ *in* $\mathcal{M}_t$ **do**
        $u_i \leftarrow \nabla_{x_i} \ell(f(x_i; \theta), y_i)$
        $\delta_i \leftarrow g(x_i, u_i; \phi)$
        Generate perturbation by $g$
    $\theta \leftarrow \theta - \alpha_1 \frac{1}{|\mathcal{M}_t|} \sum_{i \in \mathcal{M}_t} \nabla_\theta \ell(f(x_i + \delta_i; \theta), y_i)$
    Update $\theta$ over adversarial data $\{(x_i + \delta_i, y_i)\}_{i \in \mathcal{M}_t}$
    $\phi \leftarrow \phi + \alpha_2 \frac{1}{|\mathcal{M}_t|} \sum_{i \in \mathcal{M}_t} \nabla_\phi \ell(f(x_i + \delta_i; \theta), y_i)$
    Update $\phi$ over adversarial data $\{(x_i + \delta_i, y_i)\}_{i \in \mathcal{M}_t}$

---

Under this framework, the architecture of the attacker network $g$ can be very flexible. We can choose different $\mathcal{A}(x, y, \theta)$ as the input and also mimic multi-step gradient algorithms. For example, we can simply choose

$$\mathcal{A}(x, y, \theta) = x \quad \text{or} \quad \mathcal{A}(x, y, \theta) = [x, \nabla_x \ell(f(x; \theta), y)],$$

where $\nabla_x \ell(f(x_i; \theta), y_i)$ is the gradient w.r.t. $x_i$ obtained by backpropagation. Here we provide the following three examples:

**Naive Attacker Network.** This is the simplest example of our attacker network we can imagine, which takes the original image $x_i$ as the input, i.e.,

$$\mathcal{A}(x_i, y_i, \theta) = x_i \quad \text{and} \quad \delta_i = g(x_i; \phi).$$

Under this setting, L2L training is similar to GAN training. The major difference is that the generator in GAN yields the synthetic data by transforming the random noises, while the naive attacker network generates the adversarial perturbations by transforming the training samples.

**Gradient Attacker Network.** Motivated by hand-designed methods, e.g., FGSM and PGM, we design an attacker which takes the gradient information into computation. Specifically, we concatenate image $x_i$ and gradient $\nabla_x \ell(f(x_i; \theta), y_i)$ from backpropagation as the input of the attacker $g$, i.e.,

$$\mathcal{A}(x_i, y_i, \theta) = \left[x_i, \nabla_x \ell(f(x_i; \theta), y_i)\right] \quad \text{and} \quad \delta_i = g\left(x_i, \nabla_x \ell(f(x_i; \theta), y_i); \phi\right).$$

Since more information is provided, we expect the attacker network to be more efficient to learn and meanwhile yield more powerful adversarial perturbations.

**Multi-Step Gradient Attacker Network.** We adapt the RNN to mimic a multi-step gradient update. Specifically, we use the gradient attacker network $g$ as the cell of RNN. These networks share the same parameter $\phi$. Figure 1 illustrates one step in the multi-step gradient attacker network. As we mentioned earlier, the number of layers/iterations in the RNN for modeling algorithms cannot be large so as to avoid significant computational burden in backpropagation. Here we focus on a length-two RNN to mimic a two-step gradient update. Therefore, the corresponding perturbation becomes:

$$\delta_i = \Pi_{\mathcal{B}(\epsilon)}\big(\delta_i^{(0)} + g\big(x_i + \delta_i^{(0)}, \nabla_x \ell(f(x_i + \delta_i^{(0)}, y_i; \theta); \phi)\big)\big),$$

where $\delta_i^{(0)} = g\big(x_i, \nabla_x \ell(f(x_i, y_i; \theta); \phi)\big)$.

Figure 2 illustrates how to solve problem equation 3 by the gradient attacker network. As can be seen, we jointly train two networks: one classifier and one attacker. The first forward pass is used to obtain gradient w.r.t. the classification loss over the clean data. The second forward pass is used to generate perturbation $\delta_i$ using attacker $g$. The third forward pass is used to calculate the adversarial loss $\ell$ in equation 3.

Since our gradient attacker network only needs one backpropagation to query gradient, it amortizes the adversarial training cost, which leads to better computational efficiency. The corresponding procedure of L2L is shown in Algorithm 2.

## 3 EXPERIMENTS

To demonstrate the effectiveness and efficiency of our methods, we conduct numerical experiments on CIFAR-10, CIFAR-100, and SVHN datasets. We compare our methods with FGSM training and PGM training, and evaluate the robustness of deep neural networks models under both black-box and white-box settings. All experiments are done in PyTorch with one NVIDIA 1080 Ti GPU. We choose all hyperparameters by grid search.

For simplicity, we denote Plain Net as the classifier network trained over clean data only, FGSM Net and PGM Net as the classifiers with FGSM training and PGM training respectively, and Naive L2L, Grad L2L, and 2-Step L2L as the classifiers using L2L training with a naive attacker network, a gradient attacker network and a length-two RNN attacker respectively.

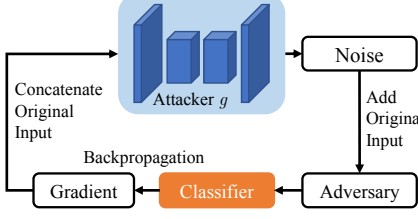

Figure 1: *The illustration of one step in the multi-step gradient attacker network.*

### 3.1 GENERAL SETTING

**Classifier Network.** All experiments adopt a 32-layer Wide Residual Network (WRN-4-32, Zagoruyko & Komodakis (2016)) as the classifier network. A pre-trained Plain Net is used as the initial classifier in the adversarial training. For training a Plain Net over SVHN dataset, we use the stochastic gradient descent (SGD) algorithm with Polyak's momentum (the momentum parameter is 0.9) and choose the step size as 0.1 for 160 epochs. The Plain Nets for CIFAR-10 and CIFAR-100 datasets are obtained by the same training procedure as Zagoruyko & Komodakis (2016). We use softmax entropy as the loss function $\ell$ for all experiments.

**Attacker Network.** We investigate two different attacker architectures: a slim network and a wide network shown in Tables 1. In the slim network, the second convolutional layer downsamples tensors in height and width, while the second last deconvolutional layer upsamples tensors. On the opposite, the wide network keeps the height and width of the intermediate tensors as the original input is. Although the slim network is computationally cheap, due to the downsampling, such an architecture loses some information of input. Thus, inspired by residual learning

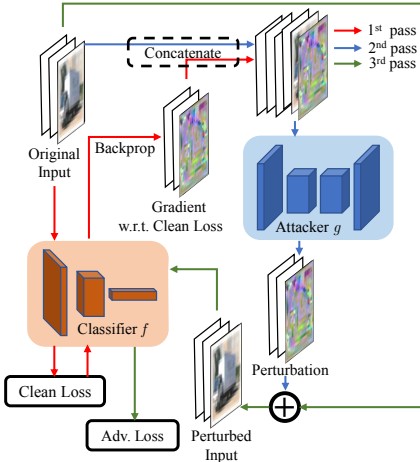

Figure 2: *The architecture of L2L adversarial training with gradient attacker network.*

in He et al. (2016), we use a skip layer connection to ease the training of the slim network. Specifically, the last layer takes the concatenation of $\mathcal{A}(x, y, \theta)$ and the output of the second last layer as input.

Table 1: *Attacker Network Architectures.* $k, c, s, p$ *denote the kernel size, output channels, stride and padding of the convolutional layers. Batch Normalization (BN) and ReLU activation are applied when specified. ResBlocks use the same structure as the generator in Miyato et al. (2018).*

| Slim Attacker Network Architecture | | |
| --- | --- | --- |
| Conv: | $[k = 3 \times 3, c = 128, s = 1, p = 1]$, BN+ReLU | |
| ResBlocks: | [channel = 256] | |
| ResBlocks: | [channel = 128] | BN |
| DeConv: | $[k = 4 \times 4, c = 16, s = 2, p = 1]$, | BN+ReLU |
| Conv: | $[k = 3 \times 3, c = 3, s = 1, p = 1]$, | tanh |

| Wide Attacker Network Architecture | |
| --- | --- |
| Conv: | $[k = 3 \times 3, c = 64, s = 1, p = 1]$, BN+ReLU |
| ResBlocks: | [channel = 128] |
| ResBlocks: | [channel = 256] |
| ResBlocks: | [channel = 128] |
| ResBlocks: | [channel = 64], BN |
| Conv: | $[k = 3 \times 3, c = 3, s = 1, p = 1]$, tanh |

**White-box and Black-box.** Under the white-box setting, attackers are able to access all parameters of target models and generate adversarial examples based on the target models; While under the black-box setting, accessing parameters is prohibited. Therefore, we adopt the standard transfer attack method from Liu et al. (2016). Specifically, we first train a surrogate model with the same architecture of the target model but a different random seed, and then attackers generate adversarial examples to attack the target model by querying gradients from the surrogate model.

**Remark 1.** *Under our black-box setting, the attack highly relies on the transferability, which is the property that the adversarial examples of one model are likely to fool other models. The transferred attack is very unstable, and often has a large variation in its effectiveness. Therefore, the results of the black-box setting might not be reliable, and we mainly focus on the results of the white-box setting. Due to the space limit, we only present the results of the black-box setting over CIFAR-10, and leave the results over SVHN and CIFAR-100 in Appendix A.*

**PGM Attack.** We use a 10-iteration PGM with a step size $\eta = 0.01$. Moreover, we adopt the same setting for training PGM net for all experiments.

**CW Attack.** Here, we briefly describe the CW attack under our setting. CW attack aims to find the-least norm perturbation that is able to fool the classifier $f$. Specifically, given one sample $x$ and its corresponding label $y$, CW attack solves the following optimization problem:

$$\delta^* = \arg\min_{\delta \in \mathcal{B}(\epsilon)} ||\delta||_\infty + c \cdot \mathcal{R}(f(x + \delta), y), \tag{4}$$

where $c$ is a tuning parameter and $\mathcal{R}(\cdot, \cdot) = \max_{t \neq y} \left( (f(x + \delta)_t - f(x + \delta)_y)_+ \right)$ penalizes those correctly classified perturbed data $x + \delta$. For solving equation 4, we adopt the update rule in Paszke et al. (2017), and set the maximum number of iterations as 100. The parameter $c$ is automatically tuned by the update rule. For each sample, our algorithm starts at 0 and stops once the perturbation successfully fools the classifier, or the maximum number of iterations is reached. Note that CW attack is only designed for white-box setting.

**Remark 2.** *CW and PGM attacks are sufficient to evaluate the robustness of neural networks, since CW attack has been shown to be close to the optimal attack Carlini et al. (2017). Moreover, we empirically find out that the results of CW attack are similar to those of PGM attack. We only evaluate the robustness against CW attack once due to its high computational cost.*

**Robustness Evaluation.** We evaluate the robustness of neural networks under both white-box and black-box settings with FGSM, PGM, and CW attacks. All reported results under the white-box setting, except CW attack, are averaged over 5 runs with different random seeds. Since the transferred attack is often not effective, we only present one result under the black-box setting to demonstrate the robustness of our models. For the maximum perturbation magnitude, we set $\epsilon = 0.03$ over CIFAR-10 and CIFAR-100, and $\epsilon = 0.05$ over SVHN. Moreover, for CIFAR-10, we also evaluate the robustness by taking different number of iterations for PGM attack with $\epsilon = 0.03$ shown in Figure 3 and different perturbation magnitudes shown in Figure 4.

### 3.2 CIFAR-10 AND CIFAR-100 DATASETS

**L2L.** To update the classifier's parameter $\theta$, we use the SGD algorithm with Polyak's momentum (the momentum parameter is $0.9$) and weight decay (the parameter is $0.0001$). We set the step size as $0.1$ for the first 30 epochs and $0.01$ for the next 10 epochs. Since the training starts from a pre-trained Plain Net, 40 epochs are sufficient for the adversarial training to converge. We use the same configuration to update the attacker's parameter $\phi$.

**FGSM Net and PGM Net.** We use the SGD algorithm to update $\theta$ for FGSM and PGM training. Different from the L2L using the step size annealing, here we use a fixed step size $0.01$, since we find that the step size annealing procedure hurts both FGSM and PGM training. Besides updating the classifier's parameter $\theta$ over adversarial samples for robustifying the classifier $f$, we also update $\theta$ over clean for keeping the accuracy on clean data as Kurakin et al. (2016) suggests. Without this trick, the accuracy over clean drops significantly for PGM Net and FGSM Net. Moreover, for PGM training, we adopt PGM10 with a step size $\eta = 0.01$ in equation 2, which yields sufficiently strong perturbations in practice.

Table 2: *Results of the white-box setting over CIFAR. (W) denotes the wide attacker network; (S) denotes the slim attacker network. Standard deviations are presented in brackets.*

| | CIFAR-10 | | | | CIFAR-100 | | | |
|---|---|---|---|---|---|---|---|---|
| | Clean | FGSM | PGM10 | CW | Clean | FGSM | PGM10 | CW |
| Plain Net | 94.49 | 23.51 | 0.00 | 0.00 | 76.10 | 9.49 | 0.13 | 0.00 |
| | (0.49) | (5.15) | – | – | (0.58) | (0.44) | (0.06) | – |
| FGSM Net | 92.44 | **76.25** | 2.90 | 0.18 | 69.39 | **52.98** | 0.68 | 0.00 |
| | (0.83) | (7.87) | (1.26) | (0.14) | (1.97) | (4.46) | (0.13) | – |
| PGM Net | 85.92 | 52.25 | 40.69 | 50.62 | 61.65 | 24.64 | 19.46 | 22.43 |
| | (0.62) | (1.20) | (0.64) | (0.56) | (1.71) | (0.44) | (0.31) | (0.31) |
| Naive L2L (S) | 94.41 | 28.44 | 0.01 | 0.00 | 75.27 | 8.47 | 0.05 | 0.00 |
| | (0.45) | (9.64) | – | – | (0.50) | (0.27) | (0.01) | – |
| Grad L2L (S) | 83.25 | 50.99 | 39.33 | – | 61.13 | 25.08 | 18.73 | – |
| | (1.02) | (0.65) | (1.04) | – | (0.34) | (0.27) | (0.48) | – |
| 2-Step L2L (S) | 75.36 | 60.19 | 46.42 | 40.82 | 60.23 | 25.92 | 20.63 | 22.70 |
| | (0.08) | (0.89) | (1.36) | (2.89) | (0.21) | (0.27) | (0.49) | (0.92) |
| Naive L2L (W) | 88.26 | 13.80 | 0.00 | 0.02 | 63.52 | 5.86 | 0.26 | 0.03 |
| | (0.85) | (1.64) | – | – | (0.26) | (0.25) | (0.08) | – |
| Grad L2L (W) | 86.92 | 60.42 | 47.90 | **53.15** | 62.43 | 34.23 | 25.92 | 28.38 |
| | (0.35) | (1.13) | (0.46) | (0.58) | (0.20) | (0.32) | (0.31) | (0.20) |
| 2-Step L2L (W) | 71.65 | 56.14 | **51.47** | 49.92 | 61.44 | 32.30 | **29.63** | **30.29** |
| | (0.70) | (0.59) | (0.14) | (0.15) | (0.13) | (0.52) | (0.50) | (0.54) |

Table 2 shows the results of all methods over CIFAR-10 and CIFAR-100 under the white-box setting [2]. As can be seen, Grad L2L and 2-Step L2L with slim attacker have comparable performance as PGM training, and Grad/2-step L2L with wide attacker significantly outperforms the PGM training. However, without taking gradient information when training the attacker, the Naive L2L is vulnerable to all the adversarial attacks. For FGSM attack, FGSM Net is the most robust since FGSM has the label leaking issue (Kurakin et al., 2016).

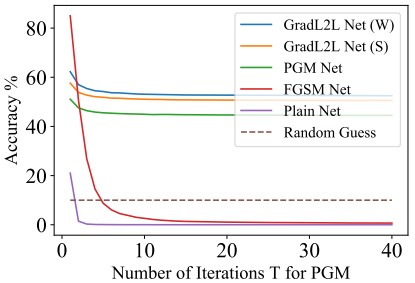

Figure 3: *The accuracy over PGM adversarial examples over CIFAR-10 with different number of iterations and $\epsilon = 0.03$.*

Besides, Figures 3 and 4 present the accuracy over FGSM and PGM adversarial data with different perturbation magnitudes and different number of iterations. As can be seen, Grad L2L is more robust than PGM Net; Both Grad L2L (S) and (W) outperform the FGSM and PGM Net with different perturbation magnitudes; We also compare the running time of all methods for one epoch over CIFAR-10 shown in Table 4. As can be seen, Grad L2L is comparable with FGSM Net. 2-Step L2L is slower than Grad L2L, but faster than PGM Net.

---

[2]Our results of PGM Net over CIFAR-10 match the results in Madry et al. (2017). For low accuracy and CW attack, we do not present the standard deviation.

Table 3: *Results of the black-box setting over CIFAR-10. We evaluate L2L methods with slim attacker networks.*

| Surrogate | Plain Net | | FGSM Net | | PGM Net | |
|---|---|---|---|---|---|---|
| | FGSM | PGM10 | FGSM | PGM10 | FGSM | PGM10 |
| **CIFAR-10** | | | | | | |
| Plain Net | 40.03 | 5.60 | 74.42 | 75.25 | 67.37 | 65.92 |
| FGSM Net | 79.20 | 85.02 | **89.90** | 80.40 | 64.28 | 63.89 |
| PGM Net | 83.80 | 84.73 | 84.33 | 85.29 | 67.05 | 65.54 |
| Naive L2L | 45.52 | 25.95 | 83.99 | 77.94 | 68.14 | 67.13 |
| Grad L2L | **86.10** | 86.87 | 87.93 | **88.01** | **71.15** | **69.95** |
| 2-Step L2L | 85.83 | **87.10** | 86.51 | 87.60 | 70.58 | 69.38 |

Table 4: *Running time for one epoch over CIFAR-10.*

| Plain Net | FGSM Net | PGM Net |
|---|---|---|
| $36.68 \pm 0.36$ s | $104.87 \pm 0.51$ s | $421.56 \pm 0.89$ s |
| Naive L2L (S) | Grad L2L (S) | 2-Step L2L (S) |
| $117.85 \pm 0.54$ s | $120.48 \pm 0.46$ s | $230.56 \pm 0.72$ s |
| Naive L2L (W) | Grad L2L (W) | 2-Step L2L (W) |
| $147.63 \pm 0.49$ s | $152.11 \pm 0.61$ s | $290.61 \pm 0.93$ s |

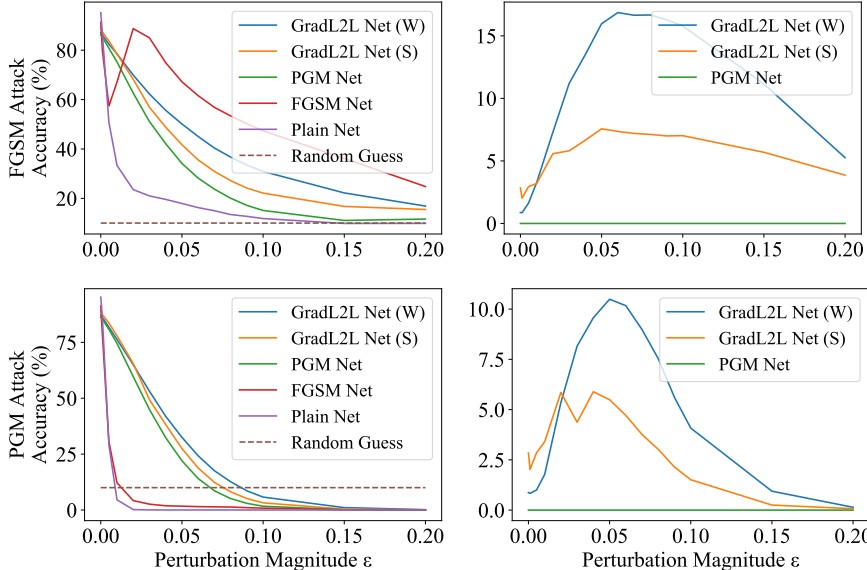

Figure 4: *Illustrative examples of performances with different maximum perturbation magnitudes over CIFAR-10 under FGSM and PGM attacks. Left figures show the accuracy over perturbed examples for all methods; Right figures show the difference between Grad L2L and PGM Net (treat the results of PGM Net as the base).*

### 3.3 SVHN DATASET

**L2L.** To update $\theta$, we use the Adam algorithm (the hyperparameters are $\beta_1 = 0.9$ and $\beta_2 = 0.99$) and weight decay (the parameter is $0.0001$) (Kingma & Ba, 2014). We set the step size as $0.0002$ for the first 30 epochs and $0.00002$ for the next 10 epochs. We train the models for 40 epochs, which is the same as the training progress of CIFAR-10 and CIFAR-100. We also use the Adam algorithm to update the attacker's parameter $\phi$ with the same hyperparameters, but a fixed step size $0.001$.

Table 5 presents the results under the white-box setting over SVHN. As can be seen, similar as in CIFAR datasets, FGSM training achieves highest accuracy against FGSM attack. Our 2-step L2L training achieves significantly higher accuracy against PGM attack, compared to PGM training. And the 2-step L2L method with wide attacker architecture has comparable performance as PGM training under CW attack. We conjecture the reason why the results of CW attack is not as good as in CIFAR datasets is because that CW attack adopts a loss function different from softmax entropy. Such a loss measures the margin of multi-class classification problem, which yields a different gradient (Crammer & Singer, 2001).

Table 5: *Experiments under the white-box setting on SVHN.*

|  | Clean | FGSM | PGM10 | CW |
|---|---|---|---|---|
| Plain Net | 96.11 | 19.62 | 0.05 | 0.07 |
|  | (0.08) | (2.07) | (0.03) | (0.05) |
| FGSM Net | 93.60 | **93.03** | 0.16 | 0 |
|  | (1.12) | (3.00) | – | – |
| PGM Net | 89.98 | 50.93 | 31.98 | **31.70** |
|  | (1.14) | (0.96) | (0.79) | (0.43) |
| Naive L2L (S) | 93.87 | 61.33 | 0.00 | 0.00 |
|  | (0.77) | (4.13) | – | – |
| Grad L2L (S) | 87.97 | 45.24 | 26.84 | 12.23 |
|  | (0.20) | (0.30) | (0.32) | (0.29) |
| 2-Step L2L (S) | 91.17 | 70.05 | **47.58** | 24.98 |
|  | (0.27) | (0.68) | (0.52) | (1.16) |
| Naive L2L (W) | 96.53 | 49.31 | 0.05 | 0.00 |
|  | (0.09) | (2.53) | (0.04) | – |
| Grad L2L (W) | 92.62 | 79.52 | 35.63 | 4.43 |
|  | (0.36) | (0.61) | (2.28) | (0.55) |
| 2-Step L2L (W) | 88.69 | 62.70 | 44.44 | 27.98 |
|  | (0.94) | (1.66) | (0.98) | (1.13) |

## 3.4 VISUALIZING ADVERSARIAL EXAMPLES

Here we present some adversarial examples of different defense models over three datasets. Figure 5 shows the FGSM and PGM perturbations of FGSM Net, PGM Net, and Grad L2L for an airplane, an apple, and a digit 5.

For the airplane and the apple, all FGSM Nets, PGM Nets, and Grad L2Ls defense FGSM attack successfully; While only FGSM Nets are fooled by PGM attack. As can be seen, the FGSM perturbations of three networks over the airplane are similar, and a similar phenomenon happens for the perturbations of the apple. The corresponding PGM perturbations, however, are very different: PGM perturbations of FGSM Net are so scattered that they look like random noises; While for PGM Net and Grad L2L, the perturbations still have certain patterns and we are able to recognize their shapes. As can be seen, PGM perturbation of the Grad L2L is similar with that of the PGM Net. This further supports our conjecture that the attacker in Grad L2L is able to learn some high order information with one step gradient information.

For the digit 5, all three networks defense FGSM attack successfully; While only Grad L2L defenses PGM attack. FGSM Net and PGM Net recognize their corresponding PGM adversarial examples as 6 and 1 respectively. As can be seen, a maximum perturbation magnitude of 0.05 for this simple dataset is large enough to generate adversarial samples to fool human beings. For example, the perturbed data of PGM attack on FGSM Net in Figure 5 is recognized as 6 by human. Moreover, the PGM perturbation of Grad L2L, however, is less destructive such that both human and Grad L2L can correctly classify the perturbed data. In this case, Grad L2L successfully learns to defense by making the gradient less informative, i.e., making gradient obfuscated Athalye et al. (2018).

## 4 DISCUSSIONS

We first discuss a few benefits of our neural network approach:

• Our attacker network $g(\mathcal{A}(x, y, \theta); \phi)$ is capable of yielding strong adversarial perturbations, since the neural network has been known to be powerful in function approximation. We generate the adversarial perturbations for all samples using the same attacker network. Therefore, the attacker network essentially learns some common structures across all samples, which help yield stronger perturbations;

• The attacker networks in our experiments are actually overparametrized. Overparametrization has been conjectured to ease the training of deep neural networks. We believe that similar phenomena happen to our attacker network, and ease the adversarial training.

We then discuss a few closely related works:

• Dai et al. (2016) leverage the Fenchel duality and feature space embedding technique, and then convert the learning conditional distribution problem to a minmax problem. This approach is quite similar to our naive attacker network. These two approaches, however, lack the primal information. In contrast, our gradient attacker network takes the gradient information of the primal variable into consideration, and achieves good results.

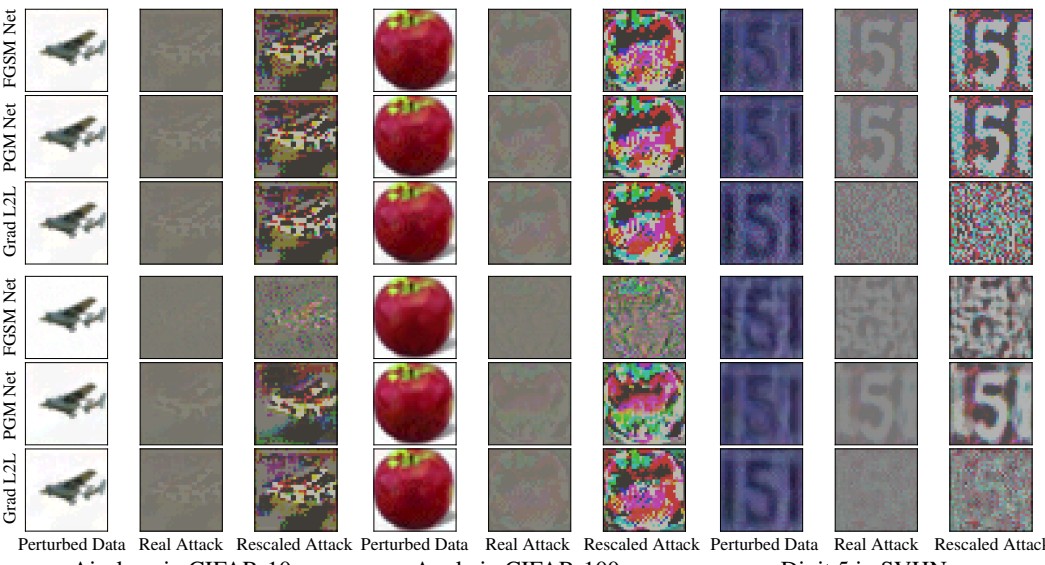

Figure 5: *Illustrative examples of FGSM and PGM perturbations for FGSM Net, PGM Net, and Grad L2L. Images in the top three layers are obtained from FGSM attack; Those in the bottom three layers are obtained from PGM attack. For the airplane and the apple, all networks defense FGSM attack, and FGSM Net fails against PGM attack; For the digit 5, all networks defense FGSM attack, but only Grad L2L defenses PGM attack.*

• Goodfellow et al. (2014a) propose the GAN, which is very similar to our L2L framework. Both GAN and L2L contain one generator network and one classifier network, and jointly train these two networks. There are two major difference between GAN and our framework: (1) GAN aims to transform the random noises to the synthetic data which is similar to the training examples, while ours targets on transforming the training examples to the adversarial examples for robustifying the classifier; Our generator network does not only take the training examples (analogous to the random noise in GAN) as the input, but also exploits the gradient information of the objective function, since it essentially represents an optimization algorithm.

The training procedure of these two, however, are quite similar. We adopt some tricks from GAN training to our framework to stabilize the training process. For example, in our Grad L2L training over SVHN, we adopt the two-time scale trick (Heusel et al., 2017).

• There are some other works simply combining the GAN framework and adversarial training together. For example, Baluja & Fischer (2017) and Xiao et al. (2018) propose some ad hoc GAN-based methods to robustify neural networks. Specifically, for generating adversarial examples, they only take training examples as the input of the generator, which lacks the information of the outer mimnimization problem. Instead, our proposed L2L methods (e.g., Grad L2L, 2-step L2L) connect outer and inner problems by delivering the gradient information of the objective function to the generator. This is a very important reason for our performance gain on the benchmark datasets.

As a result, the aforementioned GAN-based methods are only robust to simple attacks, e.g., FGSM, on simple data sets, e.g., MNIST, but fail for strong attacks, e.g., PGM and CW, on complicated data sets, e.g. CIFAR, where our L2L methods achieve significantly better performance.

## 5 CONCLUSION

This paper proposes a learning-to-learn framework to solve the adversarial training, which is a minmax optimization problem. Instead of applying hand-designed algorithms for the inner problem, we learn an attacker parametrized as a neural network. Our numerical results show that our proposed methods improve the robustness of neural networks by a margin and enjoy the computational efficiency.

We remark that the nonconvex-nonconcave minmax problems are notorious for their difficulty, and most of existing algorithms are heuristic and ad hoc. Our proposed learning-to-learn framework is well structured and can be generalized to solve more complicated minmax problems. Taking our results as a start, we expect more principled and stronger follow-up work that applies learning-to-learn to solve the minmax problem.

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

# Appendix

## A    BLACK-BOX ATTACK

As we mentioned before, evaluation under black-box setting highly relies on the transferability of the adversarial samples between models. However, such transferability is not always effective. Thus we only present one result here to demonstrate the robustness of different models.

Table 6: Experiments under the black-box setting over CIFAR-100. Note that here we only evaluate L2L methods using the slim attacker network.

| Surrogate | Plain Net | | FGSM Net | | PGM Net | |
|---|---|---|---|---|---|---|
| | FGSM | PGM10 | FGSM | PGM10 | FGSM | PGM10 |
| Plain Net | 21.04 | 9.04 | 50.57 | 54.06 | 40.06 | 41.30 |
| FGSM Net | 42.87 | 50.73 | **61.68** | 44.70 | 39.34 | 40.08 |
| PGM Net | 56.63 | 58.34 | 56.99 | 57.97 | 40.19 | 39.87 |
| Naive L2L | 20.97 | 10.47 | 50.36 | 54.07 | 38.63 | 39.91 |
| Grad L2L | 57.63 | 59.62 | 59.18 | **61.26** | 41.71 | 41.15 |
| 2-Step L2L | **58.66** | **59.31** | 58.92 | 59.46 | **45.80** | **45.31** |

Table 7: Experiments under the black-box setting on SVHN. Note that here we only evaluate L2L methods using the wide attacker network.

| Surrogate | Plain Net | | FGSM Net | | PGM Net | |
|---|---|---|---|---|---|---|
| | FGSM | PGM10 | FGSM | PGM10 | FGSM | PGM10 |
| Plain Net | 21.72 | 6.94 | 41.81 | 33.13 | 56.77 | 49.41 |
| FGSM Net | 57.36 | 51.54 | 56.25 | 38.11 | 55.99 | 48.96 |
| PGM Net | **81.04** | **81.52** | **78.66** | 80.42 | 54.85 | 49.21 |
| Naive L2L | 73.02 | 42.14 | 78.11 | 59.79 | **85.31** | **61.08** |
| Grad L2L | 71.74 | 74.31 | 77.19 | 80.70 | 71.99 | 58.71 |
| 2-Step L2L | 65.78 | 74.07 | 76.13 | **82.80** | 61.69 | 54.13 |

