# OpenReview forum: "Learning to Defense by Learning to Attack"
_ICLR.cc/2019/Workshop/DeepGenStruct — DeepGenStruct 2019_

### Official Review · AnonReviewer1 · 2019-04-15
**Good paper, though of unclear relevance**

**Rating:** 3
**Confidence:** 2

**Review:**

This paper proposes a way to train image classification models to be resistant to L-infinity perturbation attacks. The idea is to simultaneously learn the classification model and an adversary model that adds L-infinity-bounded perturbations to images, in order to maximally confuse the first model. This adversary model can use not only the image itself, but also gradient information from the classification model. It can even propose a perturbation, get gradient information on that perturbation, and then propose an updated perturbation, similarly to how projected gradient descent (PGD) can take multiple gradient steps to find a perturbation.

The main results are that training in this way improves adversarial accuracy compared to PGD, while improving training speed. On CIFAR-10 with epsilon=0.03, the proposed method gets 51.5% accuracy against a PGD adversary, whereas the PGD-trained model gets 40.7% accuracy. Madry et al. (2017) report better accuracy of 47%, but the proposed method still improves upon this. Moreover, the model-based adversary is faster than PGD, as shown by faster training times (more than 2x faster to get similar accuracy as PGD, and the best model is still about 50% faster).

Overall, I believe the paper is above the acceptance threshold from a quality perspective, and likely in the top 50% of accepted papers. However, it may not be a good fit for the topic of this workshop. Technically you could argue that the adversary is synthesizing a perturbation to an image, so this is some sort of structured generation. Therefore I give an overall rating of 3, and defer to the workshop organizers regarding appropriateness.

Minor note: In Algorithm 2, I think g(x_i, u_i; \phi) should use the \mathcal{A} notation used elsewhere.

---

### Official Review · AnonReviewer2 · 2019-04-15
**Very Interesting Idea**

**Rating:** 5
**Confidence:** 2

**Review:**

This paper proposes a very interesting idea, using the learning-to-learn framework to learn an attacker. I find this idea very novel in the literature and in retrospect, very natural. Furthermore, I believe using L2L framework to this adversarial setting is very promising as we can naturally generate many samples to fit L2L framework.

The experiments also look promising. I think this is a strong paper.

---

### Decision · Program_Chairs · 2019-04-19
**Acceptance Decision**

Accept